# Integrative Cluster Analysis of Whole Hearts Reveals Proliferative Cardiomyocytes in Adult Mice

**DOI:** 10.3390/cells9051144

**Published:** 2020-05-06

**Authors:** Anne-Marie Galow, Markus Wolfien, Paula Müller, Madeleine Bartsch, Ronald M. Brunner, Andreas Hoeflich, Olaf Wolkenhauer, Robert David, Tom Goldammer

**Affiliations:** 1Institute of Genome Biology, Leibniz Institute for Farm Animal Biology (FBN), 18196 Dummerstorf, Germany; galow@fbn-dummerstorf.de (A.-M.G.); brunner@fbn-dummerstorf.de (R.M.B.); hoeflich@fbn-dummerstorf.de (A.H.); 2Department of Systems Biology and Bioinformatics, University of Rostock, 18051 Rostock, Germany; markus.wolfien@uni-rostock.de; 3Reference and Translation Center for Cardiac Stem Cell therapy (RTC), Department of Cardiac Surgery, Rostock University Medical Center, 18057 Rostock, Germany; paula.mueller@uni-rostock.de (P.M.); madeleine.bartsch@med.uni-rostock.de (M.B.); 4Department of Life, Light, and Matter of the Interdisciplinary Faculty at Rostock University, 18059 Rostock, Germany; 5Stellenbosch Institute of Advanced Study, Wallenberg Research Centre, Stellenbosch University, 7602 Stellenbosch, South Africa; 6Molecular Biology and Fish Genetics, Faculty of Agriculture and Environmental Sciences, University of Rostock, 18059 Rostock, Germany

**Keywords:** single-cell RNA-Seq, single-nucleus RNA-Seq, harmony, cytokinesis, proliferation, heart regeneration

## Abstract

The recent development and broad application of sequencing techniques at the single-cell level is generating an unprecedented amount of data. The different techniques have their individual limits, but the datasets also offer unexpected possibilities when utilized collectively. Here, we applied snRNA-seq in whole adult murine hearts from an inbred (C57BL/6NRj) and an outbred (Fzt:DU) mouse strain to directly compare the data with the publicly available scRNA-seq data of the tabula muris project. Explicitly choosing a single-nucleus approach allowed us to pin down the typical heart tissue-specific technical bias, coming up with novel insights on the mammalian heart cell composition. For our integrated dataset, cardiomyocytes, fibroblasts, and endothelial cells constituted the three main cell populations accounting for about 75% of all cells. However, their numbers severely differed between the individual datasets, with cardiomyocyte proportions ranging from about 9% in the tabula muris data to around 23% for our BL6 data, representing the prime example for cell capture technique related bias when using a conventional single-cell approach for these large cells. Most strikingly in our comparison was the discovery of a minor population of cardiomyocytes characterized by proliferation markers that could not be identified by analyzing the datasets individually. It is now widely accepted that the heart has an, albeit very restricted, regenerative potential. However there is still an ongoing debate where new cardiomyocytes arise from. Our findings support the idea that the renewal of the cardiomyocyte pool is driven by cytokinesis of resident cardiomyocytes rather than differentiation of progenitor cells. We thus provide data that can contribute to an understanding of heart cell regeneration, which is a prerequisite for future applications to enhance the process of heart repair.

## 1. Introduction

The mammalian heart is a complex organ, composed of various cell types such as cardiomyocytes, fibroblasts, endothelial cells, smooth muscle cells, blood cells, and others [1]. Many of them display significant phenotypic, functional, and genetic heterogeneity even among the same basic cell type suggesting that their individual characteristics are influenced by a variety of factors [2]. Distinct subpopulations may be crucial for the complex biological functions of specific structures within the heart.

Genome-wide transcriptome analysis has substantially improved our understanding of the regulatory networks underlying basic cardiac biology and pathophysiological processes [3,4,5,6]. However, the resolution of gene expression patterns in the adult heart has long been limited to the level of whole tissue homogenates (“bulk”), thereby inevitably going along with loss of any information on cellular origin and on cell-type-specific changes in gene expression [7].

Recent developments in RNA amplification strategies allowed the reduction of input RNA to minimal amounts, enabling genome-wide sequencing even on the single-cell level. Single-cell RNA sequencing (scRNA-seq) offers the ability to investigate transcriptomic profiles of individual cells, providing information on prevalence, heterogeneity, and gene co-expression [8]. Along with the expansion of scRNA-seq technologies and applications with increasing sensitivity, accuracy, and throughput, new bioinformatics analysis tools (e.g., Seurat, Harmony, MTGO-SC) were developed and tailored to the specific needs [9,10].

The advent of scRNA-seq paved the way for comprehensive studies on heart biology without reliance on the cell surface or genetic lineage tracing markers. Such markers are usually imperfect and may conceal heterogeneity that exists, even within populations of similar cell types [8]. The usage of transcriptome data based on the global gene expression of individual cells renders the classification of cell types particularly quantitative and data-driven, allowing to unravel such heterogeneities, reveal novel cell populations, and identify cellular hierarchies [11,12,13,14].

The computational grouping of cells with respect to their subtypes or states is achieved by using unsupervised clustering methods such as Uniform Manifold Approximation and Projection (UMAP). The comprehensive information facilitates the discovery of novel markers defining a tissue or cell type in a hypothesis-free manner as shown in experiments disclosing the cellular composition of complex tissues [14,15,16]. Single-cell transcriptomics provides a global and unbiased insight in the cellular composition of tissues and interaction of distinct cell types, both at steady-state and in dynamic processes such as development and regeneration [8].

Recently, the Tabula Muris Consortium analyzed several organs from mice using single-cell sequencing to generate a dataset controlled for age, environment, and epigenetic effects to compile a compendium of cell types, referred to as a “tabula muris” [17]. This enabled the direct comparison of cell-type composition between organs and the comparison of shared cell types across them. The compendium comprises single-cell transcriptomic data from 100,605 cells isolated from 20 organs of C57BL/6JN mice including whole hearts [17].

However, the need for viable intact single cells can be problematic for solid organs or tissues that have highly interconnected processes. The unique difficulty of dissociating the adult mammalian heart tissue without damaging constituent cells, particularly cardiomyocytes, poses an immediate technical hurdle [18]. This, in addition to their large and irregular shape, renders most high throughput single-cell techniques vulnerable to severe technical bias [19].

To enable single-cell analysis also of large cells or cells with complex morphology, robust high throughput single-nucleus RNA sequencing (snRNA-seq) methods have been developed recently [1,2,20,21,22,23,24]. Although snRNA-seq usually detects a lower number of RNA molecules than scRNA-seq, as it excludes transcripts in the cytoplasm, it holds several technical advantages when applied to tissues difficult to dissociate, such as the mammalian heart.

First, snRNA-seq bypasses the preparation of intact single-cell suspensions and thus can be used to study frozen and fixated primary tissues [20,25]. Second, snRNA-seq can minimize the bias of recovering only cells that are easily dissociated or cells of a certain size optimal for platform-specific cell capture. Moreover, it can reduce the impact of aberrant transcriptional changes induced by enzymatic dissociation [1,26]. Although differences in type and proportion of RNA between the cytosol and nucleus do exist [23], it has been shown that nuclear transcriptomes are representative of the whole cell [21,25,26].

However, there is a particularity of the heart tissue that needs to be taken into account when relying on single-nucleus sequencing. In adult mice, ~85–90% of the cardiomyocytes are binucleated [27], so that the number of nuclei does not directly reflect the number of cells. Moreover, this binucleation has implications for cell proliferation. Cardiomyocyte polyploidization was identified as a barrier to heart regeneration [28] and it was demonstrated that binucleated cardiomyocytes that reentered the cell cycle fail to complete cytokinesis [29], rendering the regeneration of heart tissue after injury deficient.

For biomedical studies, inbred mouse lines such as C57BL/6 are commonly used. However, since human populations are genetically diverse, inbred lines may only partially contribute to an understanding of the molecular basis of a given healthy or diseased phenotype. By contrast, results from outbred mice are based on a broader genetic background and thus could be considered as more robust [30,31]. Therefore, we compare the tabula muris data not only with single-nuclei data of the respective inbred strain (C57BL/6NRj) but also with an outbred strain (Fzt:DU), to assess the extent of the technical bias and reveal biological differences in an integrative analysis.

## 2. Materials and Methods

### 2.1. Animals

For the tabula muris project, five hearts from 10–15 weeks old C57BL/6JN mice (three female and two male) had been used as described from the Tabula Muris Consortium [17].

For our approach, ten 10-week old male C57BL/6NRj were shipped from Janvier Labs to the central laboratory animal husbandry of the University Medicine Rostock, where they were euthanized by cervical dislocation the next day.

Fzt:DU mice were bred inhouse by the Service Group Lab Animal Facility. Four 12-week old male mice were euthanized for the extraction of the hearts. All mice were handled in accordance with Directive 2010/63/EU on the protection of animals. A notification about the killing of vertebrates for scientific purposes as per § 4 (3) TSchG was sent to the institutional animal welfare officer according to the German animal protection laws.

### 2.2. Isolation of Single Cells and Single Nuclei

A detailed description of the heart cell isolation procedure for the tabula muris can be found in their supplementary information of the online version under “A detailed discussion of organ cell types” [17].

Single nuclei were isolated as previously described [11]. In brief, whole hearts were harvested whereby the pulmonary artery and vena cava were removed as far as possible. The hearts were then pooled for each strain before using the Nuclei PURE Prep nuclei isolation kit (Sigma-Aldrich, St. Louis, MO, USA) according to the manufacturer’s instructions. For this, tissue samples were rinsed with ice-cold PBS and minced thoroughly with a scalpel. Remaining blood was removed by rinsing with PBS before the tissue chunks were further homogenized in chilled lysis solution using a gentle MACS dissociator. Lysate samples were mixed with chilled sucrose cushion solution and layered on 1.8 M sucrose cushion solution for density gradient centrifugation for 45 min at 30.000× *g* and 4 °C. Nuclei pellets were resuspended in chilled PBS containing 1% BSA and 0.2 U/µL RNase inhibitor and cell debris were removed by using 40 µm Flowmi cell strainers. After another centrifugation for 8 min at 600× *g* and 4 °C, nuclei were resuspended in Nuclei PURE storage buffer, snap-frozen in liquid nitrogen, and stored at −80 °C until processing.

### 2.3. Single-Cell and Single-Nucleus Sequencing

Single-cell sequencing for the tabula muris project was previously described [17]. In brief, single cells were captured in droplet emulsions using the GemCode Single-Cell Instrument (10x Genomics), and scRNA-seq libraries were constructed as per the 10x Genomics protocol using GemCode Single-Cell 3′ Gel Bead and Library V2 Kit. The samples were diluted in PBS with 2% FBS to a concentration of 1.000 cells per μL. Cells were loaded in each channel with a target output of 5.000 cells per sample. All reactions were performed in the BioRad C1000 Touch Thermal cycler with 96-Deep Well Reaction Module. Amplified cDNA and final libraries were evaluated on a Fragment Analyzer using a High Sensitivity NGS Analysis Kit (Advanced Analytical). Equal volumes of 16 libraries were pooled for sequencing on the NovaSeq 6000 Sequencing System (Illumina).

Sequencing of Fzt:DU and C57BL/6NRj samples were conducted by Genewiz (GENEWIZ Germany GmbH, Leipzig, Germany). Similar to the tabula muris project, single nuclei were captured in droplet emulsions on the 10xGenomics system and sequenced on the NovaSeq 6000 Sequencing System (Illumina, San Diego, CA, USA). In contrast to the tabula muris sequencing, cells were loaded with a target output of 10,000 cells per sample and the snRNA-seq libraries were constructed using Library V3 chemistry.

### 2.4. Computational Data Analysis

Typical data processing of scRNA-seq involves quality control, normalization, confounding factor identification, dimensionality reduction, and cell-gene level analysis [32]. Preprocessing of the raw data was conducted by using the CellRanger Software (v.3.1.0) provided by 10x Genomics. The snRNA-seq fastq data files were aligned with STAR [33] (v.2.7) to the mm10 genome (Ensembl release 93) index, annotated via GTF file and grouped by barcodes and UMIs resulting in a feature-barcode matrix. Downstream analysis was performed using Seurat [34] (v.3.1.1). After following the standard pipeline of normalization, finding variable features, scaling, and dimensionality reduction by principal-component (PC) analysis, the datasets were merged in one Seurat object to correct for batch effects and allow for an integrative analysis with the upstream processing algorithm Harmony [35] (v.1.0). The Harmony correction procedure for the newly calculated embedding’s iteratively uses its original value instead of the corrected value to regress out confounder effects. Based on this approximation, the embedding correction is restricted to a linear model of the original embedding, which results in a more robust normalization [35]. The integrated dataset was then used for UMAP clustering utilizing the formerly generated Harmony embeddings.

To assign the underlying cell types of the generated clusters, we utilized several approaches accounting for the complexity of the dataset. Sets of well-known marker genes, as well as novel cell cluster markers recently identified by other groups working with single-nuclei data [22], were applied as indicated in our provided computational script. In addition, the top 100 transcripts per cluster and the identified cell cluster markers from our dataset served as reference points for the full characterization of all clusters. The experimental protocol, computational script, top 100 transcripts per cluster, and identified cell cluster markers can be retrieved from our publicly available iRhythmics FairdomHub instance (https://fairdomhub.org/studies/739). Raw snRNA-seq data is available in the Single-Cell Expression Atlas via array express (E-MTAB-8751, E-MTAB-8848).

The used raw datasets served as a basis for three independent studies that successively build upon each other:#1Study: Heart cell type composition in entire adult mammalian hearts [11]#2Study: Strain-dependent differences in cell-type composition and velocity [36]#3Study: Discovery of proliferative cardiomyocytes by integrative cluster analysis (this study)

## 3. Results

Before data integration, the single-cell data from the tabula muris project were deviating, whereas both single-nuclei datasets appeared rather similar when looking at the first two principal components (PC) (Figure 1A). This naturally differing expression is supported by the violin plot where each point is a cell in which the specific position is based on the cell embeddings determined by the PC dimension reduction technique (Figure 1B). Dataset integration with Harmony resulted in compensation of the primarily technique-based batch effects and allowed for a direct comparison of the different datasets (Figure 1C,D). Although the datasets differed not only in the technique used to generate them but also in the detected number of features, it was possible to achieve a well-suited integrated embedding for subsequent downstream analysis.

The integrated analysis included a total of 11,672 nuclei and 628 cells representing 20,258 genes in which each cell exhibits an average total expression of 1,199 genes per cell. For the UMAP-clustering, cells were colored by their initial dataset origin to visualize the contribution of different data sets to the various clusters (Figure 2A). Some clusters contained no cells of a certain dataset, other cells appeared to be unique to one dataset, but most clusters were well mixed. The analysis identified 21 distinct clusters, of which 16 were shared by all datasets (see Appendix A provided at the iRhythmics FairdomHub instance under https://fairdomhub.org/studies/739).

The top five gene markers for all clusters were visualized in a dot plot (Figure 3). An extended visualization of the top ten gene markers per cluster and the top 100 transcripts per cluster are provided online at iRhythmics FairdomHub together with the experimental protocol and the computational script (https://fairdomhub.org/studies/739).

In the integrated dataset, the main cell types comprise populations of cardiomyocytes (30.0%), fibroblasts (26.2%), and endothelial cells (20.3%) containing ~3700, ~3200, and ~2500 nuclei/cells respectively (Figure 2B). However, their numbers severely differed between the individual datasets. For example, cardiomyocyte proportions range from only about 9% in the tabula muris data to around 23% for the BL6 data, assuming a binucleation rate of 85–90%.

Cardiomyocyte populations were identified by classical marker genes such as Actn2, Tnnc1, and Actc1. Cardiomyocytes that were characterized by high levels of markers for a mature sarcomere structure such as Myomegalin and cardiac myosin binding protein c as well as markers for contractile function such as Ryr2 but could not be clearly assigned to a ventricular or atrial type, were named “mature cardiomyocytes.” Cardiomyocytes characterized by low levels of these genes were considered “immature cardiomyocytes.”

When exploring the subpopulations, we failed to detect mature cardiomyocytes in the tabula muris dataset. However, we found some differentiated ventricular and atrial cardiomyocytes with the atrial cells constituting the major population. In contrast, BL6 showed the highest proportion of mature cardiomyocytes, but less could be assigned to clearly ventricular or atrial origin when compared with the Fzt:DU data.

In total, Fzt:DU have fewer cardiomyocytes (15.5%, considering binucleation) than BL6, but remarkably an endothelial cell population was found that shows markers clearly related to cardiomyocyte function (e.g., Gja1, Atp2a2, Ttn, Ryr2, and Myh6) suggesting a trans-differentiation process from an endothelial cell-like phenotype towards a cardiomyocyte-like phenotype in the Fzt:DU hearts [11].

We discovered an additional small population of cells (0.4%; 48 cells/nuclei) located next to the immune cell populations in the UMAP embedding, suggesting related features (Figure 2B). Indeed, cells of this small population share some common markers with monocytes such as Maf, C1qa, and Lyz2 (Figure 3). However, this population also clearly displays cardiomyocyte markers in comparable expression levels to other cardiomyocyte populations (Figure 4A–C).

Surprisingly, this population is characterized by markers related to mitosis and cytokinesis, such as Marker of Proliferation Ki-67 (Mki67) as well as the kinesin-like motor proteins Centrosome-associated protein E (Cenpe) and Kinesin Family Member 23 (Kif23), that both accumulate in the G2 phase of the cell cycle. All other cardiomyocyte populations identified in this study do not express any of these markers (Figure 4D–F).

To confirm that this population does not merely undergo mitosis resulting in binucleation instead of cytokinesis, we checked for Sept7, Anln, and Aurkb, all of which encode proteins involved in the assembly of the cleavage furrow during cytokinesis [37,38]. In fact, all of these genes were expressed by the bona fide “proliferative cardiomyocytes” (Figure 5). In contrast, Anln and Aurkb expression was almost not detectable in the other cardiomyocyte populations (average expression level <0.1), and Sept7 was expressed only in low levels from immature cardiomyocytes and atrial cardiomyocytes (Figure 5).

Because of the similarities between this cluster and the monocyte population, noticeable in Figure 3, and the fact that cardiomyocytes are usually considered postmitotic [39], we needed to fully rule out that this cluster is, in fact, a proliferative monoblast population. To this end, we investigated a variety of well-known monocyte and monoblast markers (Cd74, Cd68, Emr1, Cd14, H2-Eb1, and Cd33) and found the cells of the respective cluster to be negative for all of them as visualized as violin plot in Figure 6.

To explore whether this population might reflect resident cardiac stem cells or another progenitor cell type, we checked for typical stem cell markers. While we could not detect any c-kit or Isl1-positive cells in our dataset, we did find cells expressing Cd34, Pdgfra, or Ly6a/Sca1 (Figure 7). However, these cells are mainly members of the immature cardiomyocyte cluster.

When examining Cd34 more closely, we found the average expression of this relatively unspecific marker to be comparable or even higher in other cell clusters such as the fibroblast or the endothelial cell populations, respectively (Appendix A provided online at iRhythmics FairdomHub, https://fairdomhub.org/studies/739).

For snRNA-seq there is no possibility for a viability test before sequencing due to the isolation protocol. However, in an attempt to assess the potential impact of apoptosis on our results, we checked for the expression of apoptosis-related proteins such as those of the Bcl-2 family and inhibitors of apoptosis proteins (IAPs). We could not observe any remarkably increased expression of pro-apoptotic proteins in our data. When investigating the cardiomyocyte populations, we found no pro-apoptotic p53 or Bak and levels of the anti-apoptotic Bcl-xL and Mcl1 were comparable. We observed a slightly elevated Bax expression (average expression = 1.42) in the proliferative cardiomyocytes in comparison to average expression of around 0.35 in the other cardiomyocyte populations. However, expression levels of the antiapoptotic Bcl-2 were a seven-fold increase in proliferative cardiomyocytes (average expression = 2.0 compared to around 0.29 in other cardiomyocyte populations) and expression of the IAP survivin was even 100-fold increased (average expression = 2.23 compared to around 0.02 in other cardiomyocyte populations).

## 4. Discussion

This study underlines how significantly the applied technology affects the outcome of cluster analyses and how technical limitations regarding cell capture techniques may lead to an underrepresentation of individual cell types such as cardiomyocytes due to their large cell size and irregular shape [19]. The Tabula Muris Consortium analyzed several organs from mice using single-cell sequencing to compile a compendium of cell types, referred to as a “Tabula Muris,” knowing that cardiomyocytes will be underrepresented in their heart data [17]. Here, we directly compared their single-cell RNA sequencing data with our single-nuclei sequencing approach, especially better suited for cardiomyocytes, to likewise assess the extent of technical bias and investigate the effect of such integrative cells to nuclei clustering towards cell type identification. The generated comprehensive information also facilitates the discovery of novel markers defining a tissue or cell type in a hypothesis-free manner, refines cell type subpopulations, and validates existing markers obtained from other tissues [14,15,16].

It has previously been shown that single-nuclei techniques have several advantages over single-cell sequencing, while still retaining sufficient information for cluster analysis and matching results from whole-cell RNA in neuronal cells [21]. In line with these findings, our data accurately represent the cellular composition of the adult murine heart by relying on single-nucleus RNA sequencing. Our integrative analysis of the two single nuclei and one single-cell sequencing samples required the computational integration of these datasets via Harmony. This was a first essential processing step because the datasets were assayed with two different, yet comparable approaches in which actual biological differences might be interspersed with technical differences [23,40]. The results shown in Figure 1 highlight the necessity of such an approach, because after the Harmony processing the dimension of the embeddings values was greatly improved and aligned for a proper downstream comparison.

Considerable differences became apparent after assigning the clusters to cell types and subpopulations. The most frequent cell types were cardiomyocytes, fibroblasts, and endothelial cells, but their cell numbers differed severely between the individual datasets. For example, cardiomyocyte proportions ranged from about 9% in the tabula muris data to around 23% for the BL6 data, when taking binucleation into account. Not only do total cell numbers differ, but there is also a clear overrepresentation of atrial cardiomyocytes in the tabula muris data. We attribute this to their smaller size allowing them to pass through the microfluidics easier than e.g., ventricular cardiomyocytes, thereby representing the prime example for cell capture techniques related bias.

On the other hand, our BL6 data showed the highest proportion of mature cardiomyocytes, but less could be assigned to clearly ventricular or atrial origin when compared with the tabula muris data. The reason for this is not completely understood, but a deficiency of snRNA-seq when it comes to the specification of subpopulations cannot be excluded and even seems reasonable taking into account the limitations introduced by exclusively working with nuclear RNA.

Beyond comparison, we demonstrated that the integration of datasets results in an even more informative pool enabling deeper insights also with respect to the underlying individual data sets. Interestingly, we were able to identify a minor cell population (0.4%) in the integrated dataset that could not be detected when analyzing the datasets individually. For example, only two cells of the tabula muris dataset were identified when checking for the sample origins in this cluster, thus not being sufficient to constitute an own cell cluster in the original dataset. However, the pooling of such small populations from several datasets allows for their detection and exploration after all.

Cells of the newly detected cluster are expressing cardiomyocyte markers, but do not cluster together with other cardiomyocyte populations. Although the cells are located next to the immune cell populations in the UMAP embedding and share some common features with monocytes, an affiliation to them could be excluded due to the absence of Cd33, Cd74 (Figure 6), and other typical immune cell markers.

Strikingly, this population is characterized by markers related to mitosis and cytokinesis such as marker of proliferation Ki-67 (Mki67), centrosome-associated protein E (Cenpe) and kinesin family member 23 (Kif23). All other cardiomyocyte populations identified in this study do not express any of those markers, as would be expected from cardiomyocytes, usually considered as being post-mitotic [39].

Mki67 is commonly used to detect and quantify proliferating cells since it stabilizes individual mitotic chromosomes during all phases of mitosis. CENPE is a kinesin-like motor protein that accumulates in the G2 phase of the cell cycle and Kif 23 is a component of the centralspindlin complex required for the myosin contractile ring formation during cytokinesis. CENPE and Kif23 are both markers for late phases of mitosis and cytokinesis, but they are also present at earlier stages [37,38] and therefore, as with Mk67, there are no sufficient markers for complete cell division.

It is known that cardiomyocytes can undergo mitosis, which usually results in binucleation instead of a complete cell division. Because the aforementioned genes are markers for both mitosis and cytokinesis, we checked for additional, more specific genes.

During complete cytokinesis, a contractile ring assembles, which includes a network of actin and myosin filaments as well as septins and anillin. The cleavage furrow is formed under the regulation of several kinases such as Plk1 and Aurora B thereby acting as key regulators of cytokinesis [37,38]. Genes for septin7 and anillin as well as the genes for the mitotic kinases are expressed in considerable amounts by the cells of this cluster, thereby confirming their proliferative status.

When checking for apoptosis-related factors to evaluate the impact of potential cell death during the isolation process, we found the IAP survivin to be severely upregulated in this proliferative cell cluster. Knockout of survivin was shown to lower mitotic rate without increased apoptosis and survivin-deficient cardiomyocytes displayed marked DNA polyploidy indicative of consecutive rounds of DNA replication without cell division [41]. In contrast, overexpression of survivin rescued the cell-cycle defect in cardiomyocytes in spinal muscular atrophy mouse models as seen by the proportions of cells in the G0/G1, S, and G2/M phases of the cell cycle [42]. These data indicate that survivin is not only an antiapoptotic factor but also a critical regulator of cell division, particularly in cardiomyocytes. The increased expression of survivin in the small cardiomyocyte cluster identified in our data further strengthens the belief that these cardiomyocytes are able to undergo full cytokinesis. Vice versa the transcriptome data underline the postulated role of survivin in controlling cardiomyocyte numbers.

To explore whether this population might, in fact, represent a cardiac stem cell population, we checked for typical stem cell markers such as c-kit, Isl1, Cd34, Pdgfra, or Ly6a/Sca1 (Figure 6). Except for Cd34 there was no relevant expression of any for these genes and inspecting Cd34 more closely, we found that the average expression was comparable or even higher in other cell clusters such as the fibroblasts populations or the endothelial cell populations, respectively. These data reveal no evidence for stem cell-like characteristics in proliferative cardiomyocytes.

It is now widely accepted that the heart has a regenerative potential [43]. However, it is still debated where new cardiomyocytes arise from. Our data suggest that there are cardiomyocytes that undergo cytokinesis, which is why we termed the identified cluster “proliferative cardiomyocytes.” In accordance with other studies relying on different techniques, we propose that new cardiomyocytes in adult hearts mainly arise from differentiated cardiomyocytes rather than from resident stem cells [29,44,45]. A graphical illustration of our results can be viewed in Figure 8.

Furthermore, our study provides information on the frequency of these potentially dividing cells. In comparison with other estimations in humans [44,46] the number of proliferative cardiomyocytes seems to be quite high (0.4% of all nuclei and 1.3% of all cardiomyocyte nuclei). Indeed, not all cells of the “proliferative cardiomyocytes cluster” express all of the proliferation markers. Therefore, we cannot exclude that some cells in this population have re-entered the cell cycle but did not complete karyokinesis, and cytokinesis.

Considering 50% of the cluster (compare dot size in (Figure 3)) as cardiomyocytes truly re-entering the cell cycle and approximately 50% of them completing cytokinesis, as implied by data of Bersell et al. [29], gives a more realistic approximation in our data of around 0.3% of cardiomyocyte nuclei being derived from cells which undergo complete cell division.

Although snRNA-seq bypasses the cell capture technique related bias, it still has one drawback with respect to cardiomyocytes in particular: We cannot distinguish between mononucleated and binucleated cardiomyocytes in our analysis, thus precluding the direct calculation of cardiomyocyte cell numbers and constraining us to base our evaluation on previously determined binucleation rates.

However, for proliferative cardiomyocytes there are contradicting studies about the frequency of dividing binuclear cells. Whereas most studies suggest that proliferating cardiomyocytes were predominantly mononucleated and only mononucleated cardiomyocytes could complete cytokinesis [27,29], there are also studies claiming that mono- and bi-nucleated cardiomyocytes are in fact almost equally able to proliferate [45]. In the absence of fully verified data on the proportion of mono- and bi-nuclear cells capable to proliferate, we cannot draw an exact conclusion on the actual cell count of our identified proliferative cardiomyocyte subpopulation based on the nuclei numbers.

There are highly sensitive plate-based approaches such as Smart-Seq2 [47] and CEL-Seq2 [48] that may allow for microscopic exploration of cells before the sequencing process. This would enable visual control of binucleation on the one hand and cell division processes on the other hand. However, these are low throughput systems and due to the very rare occurrence of proliferative cardiomyocytes, it would be subject to chance whether a proliferative cell could be detected per plate at all, thus rendering plate-based systems rather unsuitable for the identification of such rare cell species at present.

In a systematic comparison of single-cell RNA-sequencing methods, it was found that the high throughput system 10x Chromium had the best, consistent performance [49], which is why we relied on this technique. It allowed us to explore the distribution and features of cell populations in entire mammalian hearts. With the ongoing development of single-cell sequencing technologies, it will in the future certainly be possible to inspect cells also in high throughput systems visually thereby enabling even more comprehensive data.

## 5. Conclusions

This study illustrates how significantly the applied technology affects the outcome of cluster analyses and demonstrates how the power of cluster analysis can be increased by integrating multiple data sets.

In summary, our data support the hypothesis of other groups postulating that there is a turnover of cardiomyocytes in the postnatal heart. Moreover, our findings suggest that the renewal of the cardiomyocyte pool is driven by cytokinesis of resident cardiomyocytes rather than differentiation of progenitor cells. Our comprehensive sequencing data can contribute to an increased understanding of heart cell biology and regeneration.

## Figures and Tables

**Figure 1 cells-09-01144-f001:**
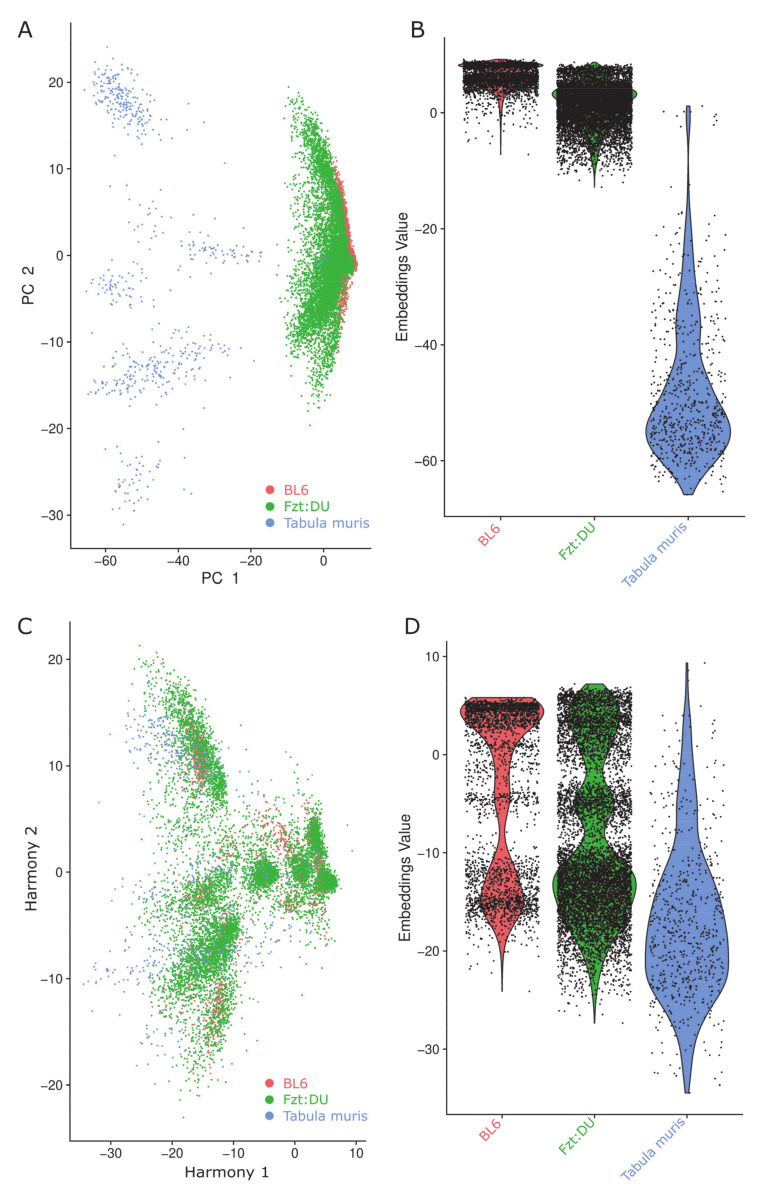
Clustering effects of the data integration with and without utilizing Harmony on the embedding in the first two principal components (PC) on the indicated datasets (BL6, red; Fzt:DU, green; Tabula muris, blue). (**A**) PCA-plot of the three investigated datasets without Harmony processing. (**B**) Violin plot of the investigated datasets without Harmony processing. (**C**) PCA-plot of the three investigated datasets with Harmony processing. (**D**) Violin plot of the three investigated datasets with Harmony processing.

**Figure 2 cells-09-01144-f002:**
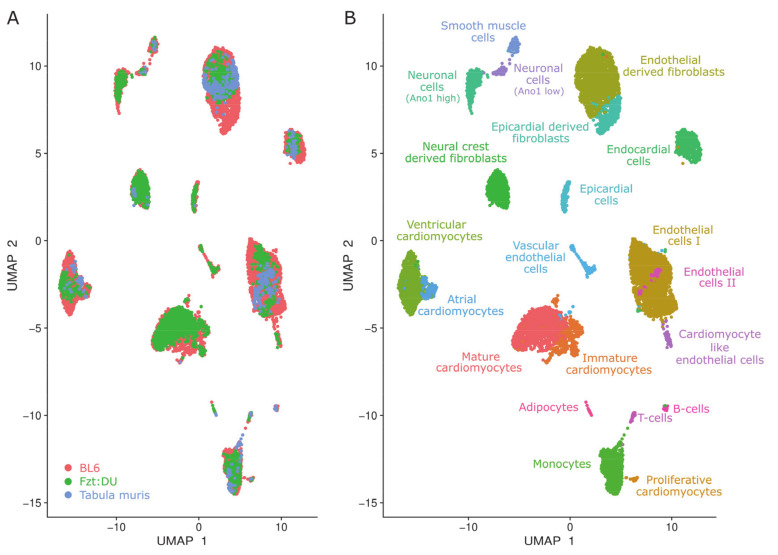
Uniform Manifold Approximation and Projection (UMAP) clustering of the integrated datasets representing the identified cellular clusters that are colored by data origin and cell type. (**A**) snRNA-seq data from whole Fzt:DU (green, 8533 nuclei) and BL6 (red, 3139 nuclei) mouse hearts were integrated with scRNA-seq data from hearts of BL6 mice of the tabula muris project (blue, 628 cells) to attain well-mixed clusters for a common downstream analysis. (**B**) 21 clusters were identified and annotated to specific cell types.

**Figure 3 cells-09-01144-f003:**
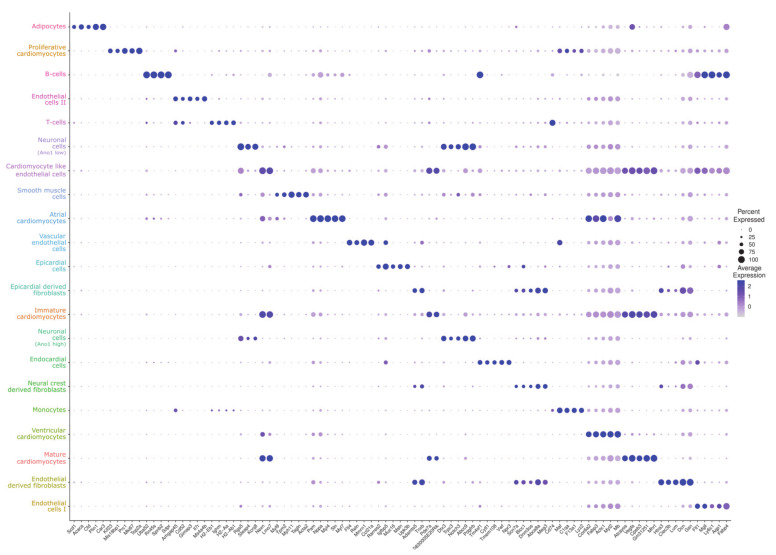
Dot plot of the top five markers for each identified cluster. For each cluster, the relative gene expression in percent is represented by the size of dots, e.g., a value of 100 means that each cell within this cell type expressed this gene. The average expression level for the listed genes is indicated by the color.

**Figure 4 cells-09-01144-f004:**
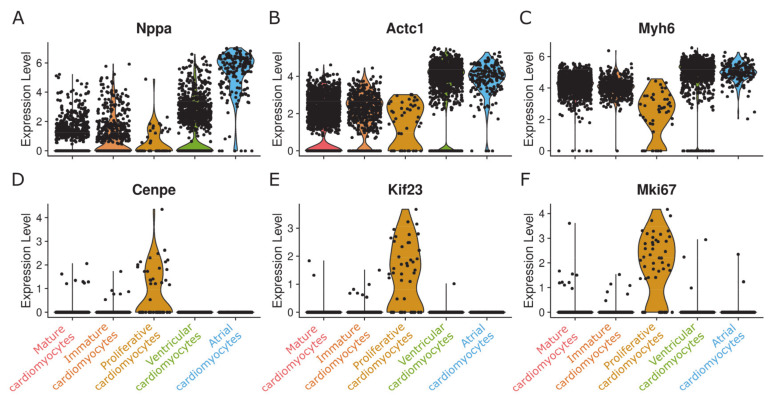
Violin plot of cardiomyocyte cell markers and markers for proliferation. (**A**–**C**) Three typical cardiomyocyte markers are expressed in comparable levels in all clusters considered as cardiomyocyte populations. (**D**–**F**) Markers for proliferation and cytokinesis are only expressed in one of the clusters, resulting in the annotation as “proliferative cardiomyocytes”.

**Figure 5 cells-09-01144-f005:**
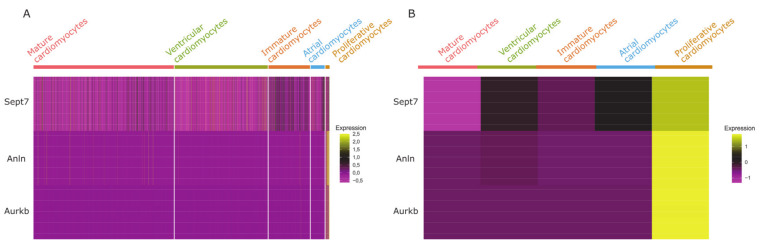
Heatmap of cytokinesis marker expression in cardiomyocyte clusters. (**A**) Expression levels of three factors involved in cleavage furrow assembly during cytokinesis are shown for each individual cell in the cardiomyocyte clusters. (**B**) Expression of cytokinesis markers averaged for each cardiomyocyte cluster shows the result more clearly, demonstrating the notable expression of cytokinesis markers exclusively in the “proliferative cardiomyocyte cluster”.

**Figure 6 cells-09-01144-f006:**
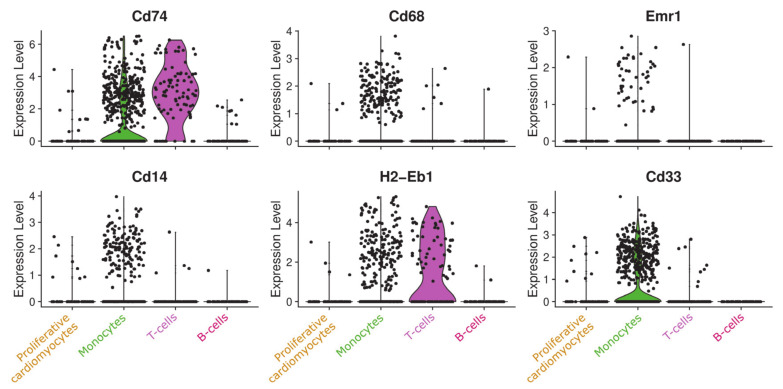
Violin plot of monoblast markers in the proliferating cardiomyocyte population and immune cells. Proliferative cardiomyocytes show no significant expression of the indicated typical monoblast markers.

**Figure 7 cells-09-01144-f007:**
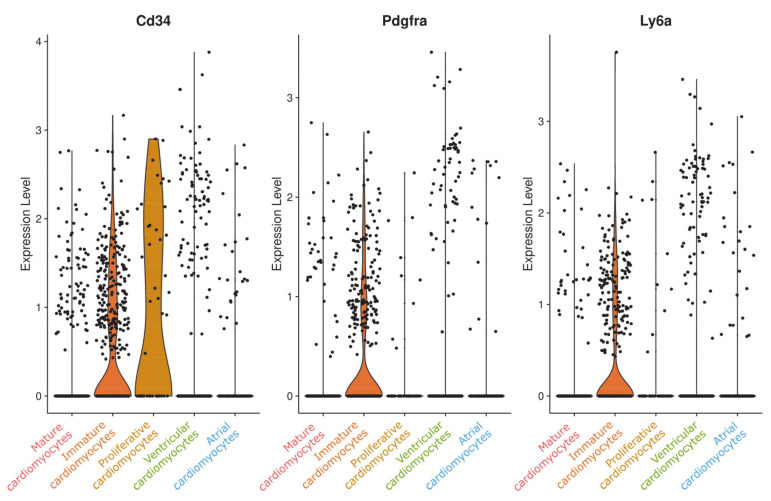
Violin plot of stem cell markers in the cardiomyocyte populations. Typical stem cell markers are expressed in low levels in immature cardiomyocytes, but not in “proliferative cardiomyocytes” except for Cd34, which showed moderate expression in several cell clusters of the integrated dataset.

**Figure 8 cells-09-01144-f008:**
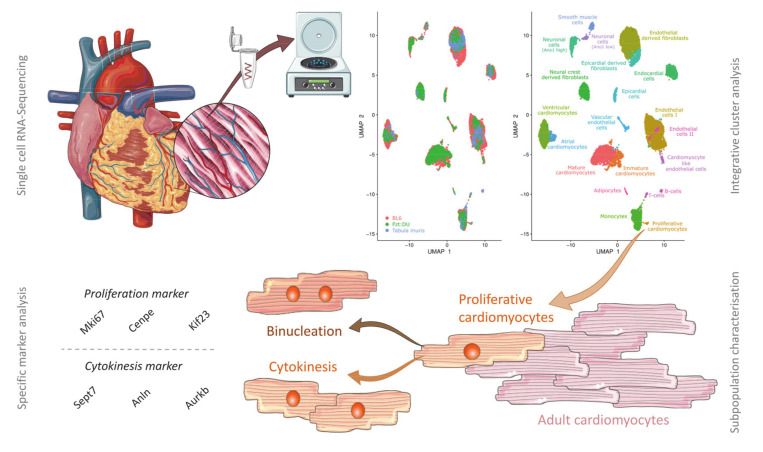
Graphical illustration of the workflow for identification of *bona fide* proliferative cardiomyocytes. Single-cell and single-nucleus sequencing data of whole adult murine hearts from inbred (C57BL/6NRj) and outbred (Fzt:DU) mouse strains were computationally integrated via Harmony. The integrated dataset allowed for novel insights on the mammalian heart cell composition revealing a minor population of cardiomyocytes characterized by proliferation markers and therefore referred to as “proliferative cardiomyocytes.” During in-depth gene expression analysis, we found additional markers specific for cytokinesis to be upregulated, suggesting that proliferative cardiomyocytes not only undergo binucleation but also complete cytokinesis. We did not detect relevant expression of any stem cell marker, thus supporting the idea that the renewal of the cardiomyocyte pool is driven by cytokinesis of resident cardiomyocytes rather than differentiation of stem cells.

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
