# Peer review of "Integrative Cluster Analysis of Whole Hearts Reveals Proliferative Cardiomyocytes in Adult Mice"

_cells, 2020, doi:10.3390/cells9051144_

Round 1
Reviewer 1 Report
In this manuscript, Galow et al. show a single-nucleus RNA sequencing (snRNA-seq) analysis of whole murine adult hearts integrated with scRNA-seq data available from the tabula muris project. By this integrative analysis, authors identified the most represented cardiac cell populations, such as fibroblasts, cardiomyocytes, endothelial cells. Among the cardiomyocyte population, the authors show a class of cardiomyocytes characterized by the presence of classical cardiac genes (Actn2, Tnnc1, Actc1) together with genes associated with sarcomere structure and contractile function. This population is named as "Mature cardiomyocytes"; cardiomyocytes that instead express low levels of these markers are named "immature cardiomyocytes". More interestingly, the authors identified a "novel" cardiomyocytes subset sharing some common markers with monocytes (Lyz2, C1qa, and Maf) but displaying also cardiomyocyte markers. Additionally, this subset of cardiac cells is characterized by mitosis- and cytokinesis-associated markers suggesting their proliferative capacity and therefore called "proliferative cardiomyocytes" and likely to be responsible for cardiac tissue regeneration. Moreover, the "proliferative cardiomyocytes" do not express any relevant marker of cardiac stem cells, such as Isl-1, c-Kit, Sca1. The authors conclude proposing that new cardiomyocytes in adult hearts arise from differentiated cardiomyocytes rather than from resident stem cells.
The work is interesting, technically solid and clearly presented, but there are some issues that need resolution.
- Are supplementary material missing? I did not find the supplementary files.
- The field of cardiac regeneration is quite controversial and highly debated and the opposite results coming from different groups make the scenario even more complicated. Please provide some biological validation of the relevance of the "proliferative cardiomyocytes in the process of cardiac regeneration. This would strengthen your work and would provide the possibility to make light on such a controversial field.
Author Response
In this manuscript, Galow et al. show a single-nucleus RNA sequencing (snRNA-seq) analysis of whole murine adult hearts integrated with scRNA-seq data available from the tabula muris project. By this integrative analysis, authors identified the most represented cardiac cell populations, such as fibroblasts, cardiomyocytes, endothelial cells. Among the cardiomyocyte population, the authors show a class of cardiomyocytes characterized by the presence of classical cardiac genes (Actn2, Tnnc1, Actc1) together with genes associated with sarcomere structure and contractile function. This population is named as "Mature cardiomyocytes"; cardiomyocytes that instead express low levels of these markers are named "immature cardiomyocytes". More interestingly, the authors identified a "novel" cardiomyocytes subset sharing some common markers with monocytes (Lyz2, C1qa, and Maf) but displaying also cardiomyocyte markers. Additionally, this subset of cardiac cells is characterized by mitosis- and cytokinesis-associated markers suggesting their proliferative capacity and therefore called "proliferative cardiomyocytes" and likely to be responsible for cardiac tissue regeneration. Moreover, the "proliferative cardiomyocytes" do not express any relevant marker of cardiac stem cells, such as Isl-1, c-Kit, Sca1. The authors conclude proposing that new cardiomyocytes in adult hearts arise from differentiated cardiomyocytes rather than from resident stem cells.
The work is interesting, technically solid and clearly presented, but there are some issues that need resolution.
- Are supplementary material missing? I did not find the supplementary files.
We apologize for this issue. The supplementary files can be accessed online from our publicly available iRhythmics FairdomHub instance (https://fairdomhub.org/studies/739). We noticed that the Hyperlink was removed in the updated word file provided by the Cells portal, but the files can also be found when browsing in FairdomHub for “proliverative cardiomyocytes”.
- The field of cardiac regeneration is quite controversial and highly debated and the opposite results coming from different groups make the scenario even more complicated. Please provide some biological validation of the relevance of the "proliferative cardiomyocytes in the process of cardiac regeneration. This would strengthen your work and would provide the possibility to make light on such a controversial field.
We appreciate the comments of the reviewer and fully agree that further research is needed to completely enlighten the mechanisms of adult heart regeneration. Unfortunately, such studies are time consuming and by far exceed the time given for this revision process. However, as also stated by Reviewer3, there are already several studies describing a significant, albeit very limited cardiomyocyte turnover in adult mammalians by relying on sophisticated fate-mapping and microscopic techniques [Bersell et al., Bergmann et al., Wang et al.]. The repetition of these experiments was not the scope of our current work, instead here we provide an additional line of evidence on a single cell transcriptomics level, supporting the results of these renowned groups.
Nevertheless, we will definitely follow the advice of this reviewer and dig deeper into this highly debated topic in our future work.
References:
Bersell, K.; Arab, S.; Haring, B.; Kühn, B. Neuregulin1/ErbB4 Signaling Induces Cardiomyocyte Proliferation and Repair of Heart Injury. Cell 2009, 138, 257–270.
Lázár, E.; Sadek, H.A.; Bergmann, O. Cardiomyocyte renewal in the human heart: insights from the fall-out. Eur. Heart J. 2017, 38, 2333–2342
Wang, W.E.; Li, L.; Xia, X.; Fu, W.; Liao, Q.; Lan, C.; Yang, D.; Chen, H.; Yue, R.; Zeng, C.; et al. Dedifferentiation, Proliferation, and Redifferentiation of Adult Mammalian Cardiomyocytes After Ischemic Injury. Circulation 2017, 136, 834–848.

Reviewer 2 Report
Comments:
This study integrated three data sets based on n=1 (one single cell RNAseq and two nuclei based snRNAseq) and analysed differencies in cell subsets. This study also seek to validate the presence of proliferating cardiomyocytes in the mouse heart. Although the use of single cell RNA sequencing of heart cells is important and relative new several studies already exist in the literature. The present study avoids the known difficulties on heart cells when performing snRNAseqs using a nuclei based approach, and obtain descriptive data that may be used in the field. However, several major concerns exist:
Points of concern:
- Mice were sacrificed the day after they arrived to the authors facility. I wonder if any of the investigated parameters can be influenced by stress? Mice have high levels of stress hormones after transfer to a new facility. The authors should consider this and argue for similar conditions between studies analyzed, and eventually investigate in their data if stress is present or not, and may harness some of the conclusions.
- During dissection, the hearts were not perfused to removed blood which is then part of the resulting scRNAseq. The authors should consider that this could be different between their analyzed dataset and skew cell type percentages between studies. The authors should look and compare blood constituents between studies.
- Details for dissection of the hearts is limited. Did the authors remove visible vessels, pericardium etc.? And were the protocols similar for data obtained in the different studies?
- Dissociation of heart tissue involved mincing etc. and many cells will die in such procedures before nuclei isolation. Is the level of dead cells before nuclei purification similar between the studies. In their sc/nRNAseq analysis the authors do not consider or analyze to exclude dead cells. Since nuclei are used, the authors cannot evaluate mitochondrial content often used for that purpose only for the scRNAseq data, but other platforms may exist for nuclei snRNAseq? Or alternatively apoptosis markers etc could be used to exclude cells that may have an aberrant gene profile impacting outcome. By claiming the existence of a small 0.4% proliferating CM population such characteristics should be analyzed to exclude the presence of dying cells.
- Although Harmonizing the datasets equals out some of the differences that indeed remain between the different studies, one may see in figure 1D, that more than half of the dots in Bl6 have Embeddings values greater than -10 and more than half dots in Tabula muris have Embeddings value less than -10. Is this not a problem for direct comparison?
- The data in figure 2 is nice and suggest though that several populations are equally distributed and present in the different datasets. However, the annotation of the cell subsets seems weak. Foremost since there is a focus on cardiomyocytes, the authors describe the use of Actn2, Tnnc1, and Actc1 to identify cardiomyocytes which is ok. But the resolution makes it impossible to check the top 5 marker genes in Figure 3 and the link for iRhythmic data is inaccessible. The authors should show the expression of these three genes in the clusters using tSNE plots.
- In relation to 6) and the overall conclusion/title a specific major concern is the annotation of the small populations referred to as proliferating cardiomyocytes. From Figure 3 despite gene names are not possible, their expression seems more to fit with a subfraction of monocytes as the authors also describe. Also, just by looking at their overall cluster location in Figure 2B, they are far away from all other types of cells referred to as cardiomyocytes. This seems weird as the authors claim expression of the cardiomyocyte markers etc. How many cells/nuclei are present in the proliferating CM cluster in Figure 2B? The authors refer to a paper on sinus node cells, which likely are very different from ventricular cardiomyocytes. Indeed, the exact procedure for identifying proliferating cardiomyocytes at present is very much unclear. Thus, the authors did not convince this reviewer on the annotations and the presence of proliferating cardiomyocytes, which is a hall mark of the paper’s conclusions. Whether the other annotations are speculative as well is unclear and was not checked by this reviewer, but should be evidenced better eventually in supplementals.
- In relation to 6-7) Instead of violin plots in figure 4-6 the authors should use tSNE cluster plots with expression of these specific genes with level distinction. They should also include plots for the Actn2, Tnnc1, and Actc1 cardiomyocyte markers. Key cytokinesis markers like Sept7, Anln, and Aurkb not shown are important and also needs to be shown in this manner. In this way the authors can better evidence that this specific small cluster indeed express the suggested genes. If not, the statements of proliferating CMs seems unsupported.
- The comparison of cell type subset percentages in Table 1 seems also a bit weak. In a study where so many parameters differ between datasets and their origin-it is hard to evidence if these differences reflect technical differences in scRNAseq of cells versus nuclei or/and other preprocessing steps and/or biology. The authors try to discuss some, but it is messy. Why not just focus on cardiomyocytes and make that discussion more detailed. Is it at all reasonable to compare between the studies and claim technical differences? To this reviewer it seems to reflect a weak experimental design that the authors base their comparison on three n=1 studies, and claim differences in technical setups. For that they should at least themselves have performed n=3 for the nuclei snRNAseq setups. The number of animals (n=10) is less important for that purpose and could have been n=3. Is the gender noted for Tabula maris???
- The origin of fibroblasts is also a bit weird in relation to biology. How did the authors decide the origin? They need to detail this or exclude and focus on CMs.
- Why can't mature cardiomyocytes in bl6 be classified into atrial cardiomyocytes and ventricular cardiomyocytes. And all the mature cardiomyocytes from Tabula muris can be classified into atrial cardiomyocytes and ventricular cardiomyocytes. Is this a deficiency of Single Nucleus RNA-Seq?
Minor points:
- when writing numbers, “,” and “.”are used in abnormal way in paragraph 2.3, the results and table 1.
- Iine265-272, this study mention the expression of Sept7, Anln, and Aurkb, I think it ’s better to use a figure here.
Author Response
Comments:
This study integrated three data sets based on n=1 (one single cell RNAseq and two nuclei based snRNAseq) and analysed differencies in cell subsets. This study also seek to validate the presence of proliferating cardiomyocytes in the mouse heart. Although the use of single cell RNA sequencing of heart cells is important and relative new several studies already exist in the literature. The present study avoids the known difficulties on heart cells when performing snRNAseqs using a nuclei based approach, and obtain descriptive data that may be used in the field. However, several major concerns exist:
Points of concern:
- Mice were sacrificed the day after they arrived to the authors facility. I wonder if any of the investigated parameters can be influenced by stress? Mice have high levels of stress hormones after transfer to a new facility. The authors should consider this and argue for similar conditions between studies analyzed, and eventually investigate in their data if stress is present or not, and may harness some of the conclusions.
We are grateful for this remark and followed the advice investigating the average expression of stress markers in our data (see table below). However, acute stress is primarily processed in the hypothalamus, therefore it is not surprising that we found only low levels of well-established stress markers like Hmgb1 and Fos in our heart cells. When checking for levels of stress hormone receptors such as glucocorticoid and mineralocorticoid receptors (Nr3c1 and Nr3c2), we found just a slightly less expressed Nr3c1 in the shipped BL6 mice compared to unshipped Fzt:DU and BL6 from the tabula muris data. Downstream targets such as Per1, Fkbp5, Lcn2, and Zfp36 were comparably low in both our snRNAseq datasets. Only for the tabula muris data elevated expression values were found for Zfp36 as well as for Fos and Hmgb1. However, this is more likely based on the different techniques used with single cell sequencing also considering the cytoplasmic RNA. Due to the limited comparability of raw gene expression levels between different RNAseq techniques, the identification of cell types in this study was based on the harmonized integrated dataset and no comparisons of individual expression values were included in the manuscript.
|
BL6 |
Fzt:DU |
Tabula muris |
|
|
Fos |
0.11 |
0.08 |
7.71 |
|
Hmgb1 |
0.95 |
1.48 |
7.93 |
|
Per1 |
0.29 |
0.41 |
0.24 |
|
Nr3c1 |
0.87 |
1.85 |
1.97 |
|
Nr3c2 |
0.16 |
0.18 |
0.05 |
|
Fkbp5 |
0.37 |
0.79 |
0.10 |
|
Lcn2 |
0.02 |
0.08 |
0.04 |
|
Zfp36 |
0.32 |
0.36 |
2.80 |
- During dissection, the hearts were not perfused to removed blood which is then part of the resulting scRNAseq. The authors should consider that this could be different between their analyzed dataset and skew cell type percentages between studies. The authors should look and compare blood constituents between studies.
Like the tabula muris consortium we relied on the well-established Langendorff-free chunk digestion method which includes a rinsing step to remove blood before addition of the lysis buffer. We changed the respective part in the materials and methods section to make this more clear for the reader (see line 132). By using the same method for all three studies we excluded this source for variation. Similar percentages of immune cells were detected in the three datasets suggesting that no substantial amounts of blood were measured.
- Details for dissection of the hearts is limited. Did the authors remove visible vessels, pericardium etc.? And were the protocols similar for data obtained in the different studies?
During dissection the pulmonary artery and vena cava were removed, while we did not attempt to remove the pericardium. According to the reviewers remark we added these details in the material and methods part (see line 129). The protocols were similar in all studies although the tabula muris consortium split the hearts in four pieces (left and right, atrium and ventricle) before performing digestion and sequencing to be able to consider differences between those areas besides the cellular composition of the whole heart. However, this could not alter their overall data for the entire heart.
- Dissociation of heart tissue involved mincing etc. and many cells will die in such procedures before nuclei isolation. Is the level of dead cells before nuclei purification similar between the studies. In their sc/nRNAseq analysis the authors do not consider or analyze to exclude dead cells. Since nuclei are used, the authors cannot evaluate mitochondrial content often used for that purpose only for the scRNAseq data, but other platforms may exist for nuclei snRNAseq? Or alternatively apoptosis markers etc could be used to exclude cells that may have an aberrant gene profile impacting outcome. By claiming the existence of a small 0.4% proliferating CM population such characteristics should be analyzed to exclude the presence of dying cells.
It was not possible to measure levels of dead cells since the minced and rinsed tissue is directly transferred into lysis buffer, which breaks up the membranes, therefore excluding any trypan or other viability test. Since the same isolation method was used in all studies, levels of dying cells during the process should be comparable. Moreover, single sequencing data of cells and nuclei have been shown to be comparable by other authors before (see Bakken et al. Single-nucleus and single-cell transcriptomes compared in matched cortical cell types. PLoS ONE 2018, and Selewa et al., Systematic Comparison of High-throughput Single-Cell and Single-Nucleus Transcriptomes during Cardiomyocyte Differentiation. bioRxiv 2019)
However, we highly appreciate the remark of the reviewer. Although evidencing apoptosis on the RNA level is slightly inapplicable, because apoptosis is induced by the activation of proteins such as death receptors or caspases already present in the cells, we did check for apoptosis related proteins like those of the Bcl-2 family and inhibitors of apoptosis proteins (IAPs) and discovered interesting findings. Whereas we found no p53 or Bak in our data and comparable levels of the anti-apoptotic Bcl-xL and Mcl1 in all populations, we observed a slightly elevated Bax expression (average expression= 1.42) in the proliferative cardiomyocytes in comparison to an average expression of around 0.35 in the other cardiomyocyte populations. However, expression levels of the anti-apoptotic Bcl-2 was 7-fold increased in proliferative cardiomyocytes (average expression=2.0 compared to around 0.29 in other cardiomyocyte populations) and expression of the IAP survivin was 100-fold increased (average expression=2.23 compared to around 0.02 in other cardiomyocyte populations). These interesting findings have been included in the manuscript (see lines 314-325 and 425-436) and we want to thank the reviewer again for putting us on this trail. In summary, the data do not imply an increased apoptosis of any cell population, but revealed another hint for the involvement of survivin in cardiomyocte proliferation.
- Although Harmonizing the datasets equals out some of the differences that indeed remain between the different studies, one may see in figure 1D, that more than half of the dots in Bl6 have Embeddings values greater than -10 and more than half dots in Tabula muris have Embeddings value less than -10. Is this not a problem for direct comparison?
It is correct that we used Harmony for projecting cells into a shared embedding in which cells group by cell type rather than dataset-specific conditions such as origin or technology used. In this way Harmony accounts for variation among technologies and donors to successfully align several rare subpopulations in an unbiased manner. The correction procedure for the embedding’s uses iteratively its original value instead of the corrected value to regress out confounder effects. In this way, the correction is restricted to a linear model of the original embedding, which results in a more robust normalization. An alternative approach would use the newly generated output embedding’s of the last iteration as input to the correction procedure. Thus, the final embedding’s would be the result of a series of linear corrections of the original embedding. While this would allow for more expressive transformations and maybe less different embedding’s, the developers of Harmony Korsunsky et al. (2019) found that in practice, this can over correct the data. Therefore, we only used the single linear correction generating already less differences among the datasets as pointed out by the reviewer. In addition to Figure 1D, we also present Figure 2A in which the cell types jointly cluster with the nuclei datasets. Taken together the above-mentioned points, we believe that a direct comparison after our Harmony correction is suitable for further downstream analysis. A description for using a single linear model was added in the manuscript (see line 165-168).
- The data in figure 2 is nice and suggest though that several populations are equally distributed and present in the different datasets. However, the annotation of the cell subsets seems weak. Foremost since there is a focus on cardiomyocytes, the authors describe the use of Actn2, Tnnc1, and Actc1 to identify cardiomyocytes which is ok. But the resolution makes it impossible to check the top 5 marker genes in Figure 3 and the link for iRhythmic data is inaccessible. The authors should show the expression of these three genes in the clusters using tSNE plots.
The annotation of cell clusters is a very challenging process since there is no reliable tool for automatic identification. Here, the identification of cell types is based on several approaches accounting for the complexity of the dataset and especially for the heterogeneity of cell populations. Sets of well-known marker genes as well as novel cell cluster markers recently identified by others were applied as provided online in the respective R-Script under “# List of marker genes for each cell type”. In addition, the top 100 transcripts per cluster were analysed using Enrichr and the identified cell cluster markers from our dataset served as further reference points for the full characterization of all clusters. We uploaded a FeaturePlot of the mentioned cardiomyocyte markers as requested by the reviewer. In fact, it shows that none of the markers is exclusive for cardiomyocytes, but all of them are expressed by individual cells of other clusters, thereby demonstrating the importance of our “multi-level-approach” for the solid identification of heterogeneous cell populations.
We now also added all figures of the manuscript as a high resolution PDF in the Fairdomhub instance of iRhythmics to enable detailed exploration of all visualizations.
- In relation to 6) and the overall conclusion/title a specific major concern is the annotation of the small populations referred to as proliferating cardiomyocytes. From Figure 3 despite gene names are not possible, their expression seems more to fit with a subfraction of monocytes as the authors also describe. Also, just by looking at their overall cluster location in Figure 2B, they are far away from all other types of cells referred to as cardiomyocytes. This seems weird as the authors claim expression of the cardiomyocyte markers etc. How many cells/nuclei are present in the proliferating CM cluster in Figure 2B? The authors refer to a paper on sinus node cells, which likely are very different from ventricular cardiomyocytes. Indeed, the exact procedure for identifying proliferating cardiomyocytes at present is very much unclear. Thus, the authors did not convince this reviewer on the annotations and the presence of proliferating cardiomyocytes, which is a hall mark of the paper’s conclusions. Whether the other annotations are speculative as well is unclear and was not checked by this reviewer, but should be evidenced better eventually in supplementals.
We understand the concerns of the reviewer and very much appreciate the input. As elucidated above, the annotation of all populations was thoroughly conducted based on several approaches. The population holds 48 cells (see online material “number of cells per cluster” and now also in the text, see line 261), rendering the amount of data smaller than for large clusters like endothelial cells or fibroblasts. However, we reviewed several aspects to exclude an incorrect annotation as discussed in the manuscript.
As the reviewer stated, exact procedures for identifying proliferating cardiomyocytes are still unclear and we are sorry that we could not convince this reviewer. However, here we provide an additional line of evidence on a single cell transcriptomics level, supporting the results of other well trusted groups describing a significant, albeit very limited cardiomyocyte turnover in adult mammalians. The reviewer is absolutely right in saying that sinus node cells are likely different from ventricular cardiomyocytes, but the several studies relying on sophisticated fate-mapping and microscopic techniques, were conducted in different species and anatomic regions [Bersell et al., Bergmann et al., Wang et al.] and are in agreement with our data from entire hearts. It is speculative to the point that we indicated the markers of the cell types to the best of our knowledge and compared it with markers from already published manuscripts, but due to the best of our knowledge there is currently no other bullet proof possibility and we believe that we took appropriate measures to avoid any incorrect annotations.
- In relation to 6-7) Instead of violin plots in figure 4-6 the authors should use tSNE cluster plots with expression of these specific genes with level distinction. They should also include plots for the Actn2, Tnnc1, and Actc1 cardiomyocyte markers. Key cytokinesis markers like Sept7, Anln, and Aurkb not shown are important and also needs to be shown in this manner. In this way the authors can better evidence that this specific small cluster indeed express the suggested genes. If not, the statements of proliferating CMs seems unsupported.
Unfortunally, tSNE is not embedded in the latest Seurat version anymore, but was completely replaced with UMAP, because it has an improved ability in preserving the global structure of the data and is faster in computation. Moreover, the cluster of proliferative cardiomyocytes is very small (48 cells) compared to the other cardiomyocyte populations, thereby rendering it extremely difficult to recognize variations of the color scale in a figure containing all cardiomyocyte clusters. We have choosen violin plots for visualization, since they allow for an easier comparison between clusters of severly varying size.
We thank the reviewer for stressing the importance of the cytokinesis markers and followed the suggestion to add a plot to visually evidence the expression of the respective genes (see new Figure 5).
- The comparison of cell type subset percentages in Table 1 seems also a bit weak. In study where so many parameters differ between datasets and their origin-it is hard to evidence if these differences reflect technical differences in scRNAseq of cells versus nuclei or/and other preprocessing steps and/or biology. The authors try to discuss some, but it is messy. Why not just focus on cardiomyocytes and make that discussion more detailed. Is it at all reasonable to compare between the studies and claim technical differences? To this reviewer it seems to reflect a weak experimental design that the authors base their comparison on three n=1 studies, and claim differences in technical setups. For that they should at least themselves have performed n=3 for the nuclei snRNAseq setups. The number of animals (n=10) is less important for that purpose and could have been n=3. Is the gender noted for Tabula maris???
There are no technical differences in the pipeline for RNAseq of cells and nuclei, since both use the exact same capture and sequencing procedure. Also the preprocessing/digestion of the heart tissue followed the same method for all sample preparations. Nevertheless, we recognize the demand of the reviewer to focus only on cardiomyocytes and followed this suggestion by removing Table1 (now provided online as supplemental data) and deleting discussions on other cell types for improved clarity.
We understand the criticism on “n=1 studies”, which are at the moment common practice in the single cell sequencing field due to the high costs and empathized by the lack of software packages fully designed to handle replicates. From a statistical point of view, replicates might help to get a better understanding for the technical noise vs. biological variation. In their technical note “Biological & Technical Variation in Single Cell Gene Expression Experiments“ 10X genomics claims a high correlation of biological replicates with a Pearson correlation of 0.97 when comparing the Gene expression profiles of neuronal tissues in two individual C57EHCV E18 mice. The pooling of ten animals in our study should render the downstream analysis robust enough to validate the existence of a certain population without validation of biological and technical noise.
The gender of mice (three female and two male) for tabula muris is noted in their supplementary information of the online version, which we originally linked in the manuscript under “A detailed discussion of organ cell types”. It is now also added in the manuscript itself (see line 116)
- The origin of fibroblasts is also a bit weird in relation to biology. How did the authors decide the origin? They need to detail this or exclude and focus on CMs.
The decision on the origin was based on markers noted in a study of Ali et al. (“Developmental Heterogeneity of Cardiac Fibroblasts Does Not Predict Pathological Proliferation and Activation”, Circulation research, 2014). Following the suggestion of the reviewer, we omitted this topic and focused on the cardiomyocytes.
- Why can't mature cardiomyocytes in bl6 be classified into atrial cardiomyocytes and ventricular cardiomyocytes. And all the mature cardiomyocytes from Tabula muris can be classified into atrial cardiomyocytes and ventricular cardiomyocytes. Is this a deficiency of Single Nucleus RNA-Seq?
We are thankful to the reviewer for drawing our attention to this point. In fact, we cannot definitely designate a specific reason for this. However, a deficiency of snRNAseq when it comes to specification of subpopulations cannot be excluded and even seems reasonable taking into account the limitations introduced by exclusively working with nuclear RNA. A short statement on this putative drawback is now included in the manuscript (see lines 358-362).
Minor points:
- when writing numbers, “,” and “.”are used in abnormal way in paragraph 2.3, the results and table 1.
We thank the reviewer for making us aware of these abnormal usages which we have corrected.
- Iine265-272, this study mention the expression of Sept7, Anln, and Aurkb, I think it ’s better to use a figure here.
As mentioned before we agree with the reviewer and added a respective figure (see new Figure 5).

Reviewer 3 Report
In the manuscript by Galow et al., the authors use snRNA seq of adult mouse hearts, and compare it to existing adult heart scRNA-seq data, to identify cardiac cell populations and clarify technical differences between single cell and single nuclei RNA-seq. The authors identify a signature for an interesting population of resident proliferative cardiomyocytes, a finding that is in agreement with recent reports in heart regeneration.
Overall, I think this is a technically sound manuscript with interesting results, and I think it is suitable and of interest for readers of "Cells". I recommend acceptance with very minor revisions.
Minor concern
Figure 3. This graph is too small and hard to read. I would suggest to expand the figure size in landscape form and use a whole page. It is also important to describe in the methodology how were the different population markers selected as descriptive of a specific population (this is easy for cardiomyocytes but gets harder for other less characterized cell types). If these methodology follows that of previous papers please provide citations.
Author Response
In the manuscript by Galow et al., the authors use snRNA seq of adult mouse hearts, and compare it to existing adult heart scRNA-seq data, to identify cardiac cell populations and clarify technical differences between single cell and single nuclei RNA-seq. The authors identify a signature for an interesting population of resident proliferative cardiomyocytes, a finding that is in agreement with recent reports in heart regeneration.
Overall, I think this is a technically sound manuscript with interesting results, and I think it is suitable and of interest for readers of "Cells". I recommend acceptance with very minor revisions.
Minor concern
Figure 3. This graph is too small and hard to read. I would suggest to expand the figure size in landscape form and use a whole page. It is also important to describe in the methodology how were the different population markers selected as descriptive of a specific population (this is easy for cardiomyocytes but gets harder for other less characterized cell types). If these methodology follows that of previous papers please provide citations.
We are grateful to the reviewer and agree that Figure 3 is not optimal in its current dimension. If the editors agree, we will be happy to expand this figure to landscape form. To accommodate the readers in any case, we additionally uploaded all figures from the manuscript as a high resolution PDF on our FairdomHub/iRhythmics instance (https://fairdomhub.org/studies/739) for an improved visualization and inspection.

Round 2
Reviewer 1 Report
I still think that biological validations to support the data shown by Galow et al. need to be shown even if time-consuming as declared bu authors in their cover letter.
Anyway, if the other two reviewers and the Editor think this might not be a critical point then it means the paper can be accepted in its current version.
Author Response
We acknoweledge the advice of Reviewer 2 very much and as we continue to explore the subject in greater depth in future studies, we will conduct biological validations as well. According to the remarks of the editor, we added another paragraph on the strengths and limitations of the used system (see line 478-501; 428-451 without "Markups") and are now positive that the paper can be accepted in its current version.